# Advances in EUS-Guided Biliary Drainage for the Management of Pancreatic Cancer

**DOI:** 10.3390/cancers17213428

**Published:** 2025-10-25

**Authors:** Thomas Lambin, Sarah Leblanc, Bertrand Napoléon

**Affiliations:** Digestive Endoscopy Unit, Hôpital Privé Jean Mermoz, Ramsay-Santé, 69008 Lyon, France; sarahleblanc34@hotmail.com (S.L.); dr.napoleon@wanadoo.fr (B.N.)

**Keywords:** pancreatic cancer, EUS-guided biliary drainage, EUS–choledochoduodenostomy, EUS–hepaticogastrostomy, lumen-apposing metal stent

## Abstract

**Simple Summary:**

Obstructive jaundice caused by pancreatic cancer remains a common and challenging clinical condition. Traditionally, biliary drainage has been performed using endoscopic retrograde cholangiopancreatography or percutaneous transhepatic biliary drainage. However, these conventional approaches are not always feasible and may be associated with significant complications. Endoscopic ultrasound-guided biliary drainage (EUS-BD) has emerged not only as a reliable rescue technique when standard methods fail but also as a promising first-line option in selected cases. This review summarizes the main EUS-BD techniques and their current role compared with conventional strategies. It also highlights recent technological innovations likely to further strengthen the role of EUS-BD in clinical practice. Additionally, common clinical scenarios in which EUS-BD can be applied are outlined, along with a practical algorithm for the management of malignant biliary obstruction. Overall, this review aims to help clinicians better understand how these techniques can improve patient outcomes and contribute to more effective multidisciplinary care.

**Abstract:**

The indications for biliary drainage in cases of pancreatic head tumors with biliary obstruction are well established. ERCP with stent placement has long been the gold standard technique, outperforming surgery or percutaneous drainage. However, in cases of distal malignant biliary obstruction, ERCP becomes more complex, increasing the risk of complications. The advent of therapeutic endoscopic ultrasound (EUS), particularly EUS–choledochoduodenostomy (EUS-CDS) and EUS–hepaticogastrostomy (EUS-HGS), has transformed the management of distal malignant biliary obstruction in the case of pancreatic cancer. EUS-CDS creates communication between the duodenum and the common bile duct. Lumen-apposing metal stents (LAMSs) simplify the procedure, offering high technical and clinical success rates and making the technique easier to perform. Nevertheless, long-term dysfunction rates remain high, necessitating careful definition of procedural indications. EUS-HGS, a more complex technique, connects dilated left bile ducts to the stomach and requires advanced expertise; it is associated with a higher rate of complications. However, its clinical efficacy and technical success are comparable to those of EUS-CDS, and it is the preferred technique in cases of duodenal obstruction or altered anatomy. European and American guidelines currently position EUS-guided biliary drainage (EUS-BD) as a second-line approach after ERCP failure or when ERCP is not feasible, but there is a growing trend toward earlier use. Other techniques are emerging, such as EUS-guided gallbladder drainage (EUS-GBD) and combining EUS-HGS with antegrade stenting, offering valuable alternatives when conventional techniques fail or are inaccessible.

## 1. Introduction

Indications for biliary drainage in cases of pancreatic head tumors causing biliary obstruction are well established: patients with cholangitis, bilirubin levels > 250 µmol/L, severe symptomatic jaundice (e.g., intense pruritus), delayed surgery, and before neoadjuvant chemotherapy in jaundiced patients. Until recently, the reference technique for biliary drainage was the placement of a stent via endoscopic retrograde cholangiopancreatography (ERCP). Various studies have demonstrated the superiority of this strategy compared to surgical treatment or percutaneous transhepatic biliary drainage (PTBD) [1]. In the case of distal malignant biliary obstruction, ERCP is more challenging and carries an increased risk of complications that may negatively impact patient prognosis [2]. The advent of therapeutic endoscopic ultrasound (EUS), including EUS–choledochoduodenostomy (EUS-CDS) and EUS–hepaticogastrostomy (EUS-HGS), has profoundly transformed the management of patients in this context. In this review, we discuss EUS-guided biliary drainage (EUS-BD) techniques and their positioning relative to traditional approaches. We also highlight recent advances that are set to further enhance the role of EUS-BD in the management of pancreatic cancer.

## 2. EUS–Choledochoduodenostomy

### 2.1. Principle

EUS-CDS involves creating communication between the duodenum and the common bile duct (CBD) to bypass the tumor-related obstruction (Figure 1). This technique has existed for many years and initially used plastic biliary stents, but they are associated with a higher risk of peritonitis and can become occluded, resulting in cholangitis and the need for repeat interventions. Subsequently, self-expanding metallic stents (SEMSs) were used to increase drainage diameter and stent patency [3]. Since their first description in 2014 [4], the use of lumen-apposing metal stents (LAMSs) for EUS-CDS has greatly simplified the procedure. A meta-analysis pooling 31 studies with 820 patients who underwent EUS-CDS with either SEMSs or LAMSs found similar performance in terms of technical and clinical success and adverse events, although LAMS placement appeared to be faster than SEMS [3]. In practice, the ease of use of LAMSs makes it preferable to SEMSs, especially since it can be performed without fluoroscopy. Several types of LAMSs are currently available, including Nagi™ (Taewoong Medical, Gimpo-si, Gyeonggi-do, Republic of Korea), Hanarostent™ Plumber (M.I. Tech, Pyeongtaek, Republic of Korea), Hot Spaxus™ (Taewoong Medical, Gimpo-si, Gyeonggi-do, Republic of Korea), and Hot Axios™ (Boston Scientific, Marlborough, MA, USA) stents. Among them, Hot Spaxus™ and Hot Axios™ stents are the most frequently employed for EUS-CDS. Comparative studies between these two devices remain limited; however, available data suggest that they offer comparable technical and clinical success rates, as well as similar profiles of adverse events when used for EUS-CDS [5,6]. These stents avoid the need for a preliminary fistulotomy thanks to their all-in-one system [4]. This device includes a stent preloaded in a deployment system with an electrocautery tip to create an anastomosis. Under endoscopic ultrasound guidance, the operator positions the scope in the duodenal bulb to visualize the dilated CBD (>15 mm, see below). The absence of cystic duct interposition is confirmed, and Doppler ensures there are no intervening vessels. With the electrocautery LAMS device, the duodenal wall is punctured, then the bile duct. Once the bile duct is accessed, the distal flange is deployed and the stent is retracted to achieve good apposition between the bile duct and duodenum. Then, the proximal flange is released. It is recommended to perform EUS-CDS in cases where the CBD measures > 12 mm for experts and >15 mm for non-experts. The presence of ascites along the puncture tract is a contraindication; in such cases, paracentesis may be required before performing the drainage procedure [7].

Some clinical and biological criteria have been described to predict the feasibility of EUS-CDS. In a study by Rimbas et al., patient age, bilirubin level, and the interval between onset of jaundice and the EUS procedure were associated with dilation of the CBD > 12 mm. Only age and bilirubin were associated with a CBD > 15 mm. However, these two criteria were not linked to the extent of dilation, but a model with bilirubin ≥ 120 µmol/L and age ≥ 70 years could predict dilation > 12 mm. This helps predict which patients are suitable candidates for EUS-CDS [8].

### 2.2. Efficacy/Adverse Events

Several meta-analyses have evaluated the technical and clinical outcomes of EUS-CDS, reporting technical success rates ranging from 93.5% to 96% and clinical success rates ranging from 88% to 96% [3,9,10,11,12,13]. A study with 52 patients analyzed factors associated with procedural success: performing the recommended technique (direct bile duct puncture without guidewire), having a CBD diameter > 15 mm, and using a 6 mm stent were associated with success in univariate analysis [14]. Another study found that there were fewer adverse events with 8 × 8 mm LAMSs compared to 6 × 8 mm [12]. The overall complication rate ranged from 5.2% to 20% [3,9,10,11,12,13]. The most frequent complications were cholangitis and cholecystitis. Other possible complications include peritonitis, bleeding, bilioperitoneum, pneumoperitoneum, stent migration, abdominal pain, and double mucosa puncture [15]. A study by Jacques et al. compared the results of EUS-CDS with LAMSs performed by experts and non-experts (<20 EUS-BD procedures) and found no significant difference in terms of procedure duration (9.9 min vs. 10.6 min; *p* = 0.85), clinical success rate (96.6% vs. 100%; *p* > 0.99), and technical success rate (82.8% vs. 95.7%; *p* = 0.21) [14]. This suggests that EUS-CDS is a technique more readily accessible to novice operators, in contrast to EUS-HGS, which requires more experience.

### 2.3. Management of Mispositioned Stent

Misposition of LAMSs is a notable complication during drainage procedures. One study [16] proposed a detailed classification of such mispositions (Table 1).

In the same study, independent risk factors for technical failure of EUS-CDS included a CBD diameter ≤ 15 mm, limited operator experience (≤10 LAMS procedures), the presence of duodenal stenosis, and the use of the guidewire technique. Failures were predominantly type 2 (41%), followed by type 1 (26%), type 5 (16%), type 3 (11%), and type 4 (6%). Endoscopic rescue varies by type of failure (Table 1): 73% of type 2 failures were corrected by guidewire placement through the LAMS catheter, followed by the deployment of a fully covered SEMS (FC-SEMS) or the immediate placement of a new LAMS. For type 1, 81% were rescued using a FC-SEMS and/or pigtails. Type 3 was manageable in 91% of cases by deploying a FC-SEMS, and type 4 was rescued in half of the cases either by performing a new EUS-CDS with FC-SEMS over a guidewire or repositioning the catheter with a new LAMS for guidewire insertion (Table 1).

Overall, the situation was rescued in 53% via guidewire placement and FC-SEMS and in 22% by positioning a new LAMS. Other approaches include ERCP, EUS-HGS, and gallbladder drainage. The overall success rate of endoscopic rescue reached 77%, reducing severe complications, but 12% of technical failures still resulted in death within 30 days [16].

### 2.4. Long-Term Outcomes

An increasingly reported issue with EUS-CDS is dysfunction, characterized by a rise in bilirubin or the onset of cholangitis, which leads to impaired biliary drainage and necessitates reintervention. A variable rate of 2.7% to 55% has been described [14,17,18,19,20,21,22,23,24,25,26,27,28,29,30].

Some of these dysfunctions may be linked to the fact that the stent is covered and that the perpendicular position of the flange to the bile duct wall makes LAMS prone to obstruction by the contralateral wall of the bile duct. Additionally, placement in the duodenal bulb, just opposite to the pylorus, and the reduced saddle size of the stent facilitate food reflux into the bile duct [31].

A classification has been described to group all causes of LAMS dysfunction (Table 2) [20]. The most common causes of dysfunction are stone/sludge impaction, followed by gastric outlet obstruction (GOO), and then food impaction. In the same study, duodenal invasion was the only independent predictor of dysfunction, with a hazard ratio (HR) of 2.7 (95% CI 1.1–6.8) [20]. The main techniques to manage dysfunction include placing double pigtail stents in the LAMS, using a balloon extractor to flush the stent, and treating digestive obstruction without intervening on the LAMS itself. Most of the time, an endoscopic reintervention is sufficient to treat dysfunction; radiologic drainage or surgery is used only in exceptional cases. Placement of a new EUS-CDS or a PTBD remains an option. In some cases, the LAMS can be used as access for a guidewire to place a SEMS via the antegrade route or perform a rendezvous technique [32].

Various methods have been described to prevent dysfunction. Among these are the systematic placement of a double pigtail or a covered SEMS after LAMS insertion. The double pigtail helps to keep the contralateral bile duct wall away. The SEMS has the same effect and redirects the stent lumen into the duodenum toward the alimentary side, allowing flow to pass alongside the stent rather than inside it. SCORPION-IIp is a prospective study evaluating the routine placement of FC-SEMS through the LAMS on 24 patients showing a dysfunction rate of 10%, which is relatively low compared to previously reported rates, though the success rate for stent placement was 83% in LAMS. It should be noted that two patients developed cholecystitis due to cystic duct obstruction [31]. The complexity of the procedure and its cost currently do not support the routine use of SEMS.

An alternative to SEMS placement is to introduce pigtails into the LAMS. Initial data did not demonstrate its efficacy; however, this was based on a retrospective, single-center study [33]. The same group then conducted a randomized, multicenter trial evaluating this approach in patients with distal malignant biliary obstruction. A total of 47 patients were randomized to the LAMS-only group and 44 to the LAMS + pigtail group (a single 7Fr pigtail was preferably positioned intrahepatically to direct bile flow), with a one-year follow-up. The rate of recurrent biliary obstruction (RBO) was lower in the LAMS + pigtail group compared to the LAMS-only group—9% versus 30%, respectively, with a relative risk (RR) of 0.30—but there were similar biliary reinterventions in both groups (12 vs. 11%). There was a longer hospital stay in the LAMS group (median difference of 4.5 days). No difference was observed in terms of clinical and technical success or adverse events, but the procedure time was shorter in the LAMS-only group (median difference of 11 min) [34].

## 3. EUS–Hepaticogastrostomy

### 3.1. Principle

This technique, first described in 2003 by Giovannini et al. [35], creates a connection between dilated left biliary ducts and the gastric cavity [36] (Figure 1). It is a complex procedure requiring a technical platform with visceral surgeons and an interventional radiologist available in the case of complications [37]. High-quality cross-sectional imaging is also a prerequisite to assess the possibilities for access to the bile ducts and any potential difficulties (ascites, portal hypertension, etc.). The first step consists of positioning the echoendoscope in the stomach to visualize the dilated left bile ducts. The bile ducts are then punctured ensuring at least 2.5–3 cm of liver tissue depth to avoid bile leakage. Biliary dilation greater than 5 mm is preferred. Bile is aspirated to confirm correct positioning and a cholangiogram is performed to map the biliary tree. A guidewire is advanced into the left bile duct and/or CBD. Over this guidewire, an anastomosis is created between the bile duct and the stomach to allow passage of the stent with a cystotome. A dedicated partially covered SEMS is then deployed. These stents have two parts: a proximal uncovered portion to avoid blocking collateral ducts, and a distal covered portion to protect the hepaticogastric anastomosis, with an anti-migration flare at the distal end. The stent is positioned so that the junction between the covered and uncovered portions aligns with the bile duct entry. The proximal stent portion is deployed within the endoscope channel, and 2–3 cm of stent should extend into the gastric lumen [7]. Contraindications include abundant ascites, hepatic lesions obstructing biliary puncture, and interposed vessels, especially in the case of portal hypertension [7,36]. As with EUS-CDS, the presence of ascites along the EUS-HGS puncture route is a contraindication and requires drainage prior to the procedure. A study conducted in centers with limited experience in EUS-BD—mostly involving EUS-HGS or EUS–hepaticojejunostomy—showed that moderate (extending to the liver surface) to severe (between the stomach and liver) ascites was a risk factor for complications [38]. In the presence of ascites, it is therefore preferable to perform the procedure in an expert center.

### 3.2. Efficacy/Adverse Events

Several meta-analyses have reported the results of EUS-HGS [39,40,41,42,43,44]. Clinical efficacy exceeds 85%, with a technical success rate above 90%. However, even with dedicated stents that facilitate the procedure, this technique remains complex and is associated with a high adverse event rate—complications have been reported in up to nearly 30% of patients. The most common complications are bile leakage, cholangitis, bleeding, pneumoperitoneum, stent migration or obstruction, and hepatic abscesses; fatalities are rare but occasionally occur [36]. The learning curve is long, and EUS-HGS should be performed by expert EUS endoscopists. One study showed that after 33 cases, the complication rate decreased from 36.4% to 20.8% and remained stable after this threshold [45]. The same study demonstrated that a biliary duct diameter less than 5 mm was associated with a 3.7-fold increased risk of failure, and an intraparenchymal stent portion exceeding 3 cm was linked to a 5.7-fold higher risk of complications [15]. Conversely, another study found that a distance under 2.5 cm correlated with a risk of biliary peritonitis with an odds ratio (OR) of 96.98 (10.12–929.12) [46]. Therefore, an intraparenchymal portion between 2.5 and 3 cm is preferable for EUS-HGS.

### 3.3. Long-Term Outcomes

Long-term outcomes of EUS have received little research attention. One study specifically assessed these outcomes in 211 patients with EUS-HGS and found a rate of 19.1%, with stent patency rates of 88.9% [84.1; 93.9], 82.2% [75.8; 89.1], 69.5% [60.2; 80.1], and 61.7% [50.7; 75.2] at 30, 90, 180, and 365 days, respectively. The leading cause of RBO was tumor ingrowth (36.8%), followed by food impaction or sludge formation (21.1%), hyperplasia at the uncovered portion of the stent (15.8%), and finally bleeding or clot formation in 7.9% of cases. The use of a partially covered SEMS was associated with a lower risk of RBO compared to FC-SEMS. Similarly, patients with distal biliary stenosis had better biliary patency (HR 0.06, *p* = 0.031) [42]. Another study assessed long-term outcomes using a long, partially covered stent in 110 patients. RBO occurred in 33% (n = 36) of cases, with a median time to RBO of 3.2 months (interquartile range, 2.0–5.9). In this study, the major cause of RBO was hyperplasia at the uncovered portion (23%). RBO did not differ by stent size, but the rate of RBO was higher in cases of prior biliary drainage (45% vs. 15% in cases with and without prior biliary drainage, respectively, *p* < 0.01). RBO due to hyperplasia was primarily managed with stent-in-stent placement through the EUS-HGS route, using either uncovered SEMSs or plastic stents. RBO due to sludge was primarily managed using a balloon sweep through the EUS-HGS route [47].

### 3.4. EUS-CDS Versus EUS-HGS

Several meta-analyses have compared EUS-HGS and EUS-CDS, generally indicating that there is no significant difference in terms of clinical and technical efficacy. Two analyses found a higher complication rate for EUS-HGS compared with EUS-CDS, with an OR of 2; the second demonstrated a shorter procedure time for EUS-CDS compared with HGS. One meta-analysis reported fewer reinterventions and less stent obstruction with EUS-CDS than with HGS [9,48,49,50,51,52]. Most meta-analyses included patients in whom EUS-CDS was performed with a SEMS, and LAMSs are superior to SEMSs for EUS-CDS [3]. In cases of distal malignant biliary obstruction associated with duodenal stenosis, EUS-HGS should be favored; it is also preferable in patients with altered anatomy (see below). The main characteristics of EUS-CDS and EUS-HGS are summarized in Table 3.

## 4. Historical EUS-BD Techniques

### 4.1. EUS-Rendezvous

The EUS-rendezvous (EUS-RDV) technique consists of puncturing the dilated bile duct with a 19G needle under endoscopic ultrasound guidance. The EUS scope is usually positioned in the duodenal bulb to access the CBD, but puncture of the dilated left bile ducts via a transgastric approach is also possible. Once punctured, a guidewire is advanced into the bile duct, then through the stricture and into the duodenum. The endoscope is then withdrawn, leaving only the guidewire in place. Next, a duodenoscope is introduced to the papilla, where the guidewire exits; it is then snared with a loop or foreign body forceps. The retrograde cannulation is then performed as usual. Alternatively, a sphincterotome can be used to cannulate the papilla, with the rendezvous guidewire in place to facilitate access [7,53].

A recent randomized trial compared papillotomy with EUS-RDV in distal malignant biliary obstruction, with 104 patients in each group. There was no significant difference in technical success or adverse events, but the rate of pancreatitis was higher in the papillotomy group compared to EUS-RDV (8.7% vs. 1.9%, *p* = 0.06). The EUS-RDV procedure time was nearly twice as long (47 min vs. 27 min) [54]. The main drawbacks of this technique are its duration and, in cases of markedly dilated bile ducts, the guidewire may coil within the duct and fail to traverse the stricture. In practice, the development of EUS-HGS and EUS-CDS using LAMSs has rendered this technique less common.

### 4.2. EUS-Antegrade Stenting

The initial principle is similar to EUS-HGS, involving puncture of a dilated left bile duct via a transgastric approach, cholangiography to map the biliary tract and localize the obstruction, and introduction of a guidewire through the malignant stricture and across the papilla into the duodenum. An anastomosis is then created between the stomach and the bile duct using either a cystotome or a dilation balloon, enabling passage of a metallic stent across the anastomosis and the stricture [7]. In a pooled analysis of several studies, the technical success rate was 92% (95% CI 87.9–95.3), with an adverse event rate of 14% (95% CI 9.3–19.9). Pancreatitis was a main concern due to ampullary stent coverage [55]. One study compared this technique to PTBD and found similar clinical and technical success rates (>90%), but the adverse event rate was more than twice as high in the PTBD group, albeit not statistically significant (11.4% vs. 27.6%, *p* = 0.119) [56].

## 5. Emerging EUS-BD Techniques

### 5.1. EUS-Guided Gallbladder Drainage for Biliary Decompression

Performing biliary drainage via EUS in cases of ERCP failure is not always feasible. This has led some to consider gallbladder drainage as an alternative to standard techniques, utilizing the cystic duct to ensure biliary drainage (Figure 1). EUS-guided gallbladder drainage (EUS-GBD) has been described for several years in cases of cholecystitis, with excellent technical and clinical results. A recent meta-analysis showed a technical success rate of 95.8% (95% CI 93.9–97.2%) and a clinical success rate of 94.3% (95% CI 92–96%), with a cholecystitis recurrence rate of 4.2% (95% CI 2.8–6.2) [57]. The main indication is cholecystitis in inoperable patients.

Three recent meta-analyses have evaluated the role of EUS-GBD in distal malignant biliary obstruction. The clinical success rate was between 85% and 89%, with 10–13% adverse events, a technical success rate between 99.2% and 100%, and a dysfunction rate of 9% [58,59,60]. Of note, most of the studies included in these meta-analyses were retrospective in expert tertiary centers, so the results should be interpreted with caution. A recent study not included in previous meta-analyses reported a similar technical success rate, but a slightly lower clinical success rate (78.9%) and a higher adverse event rate (26%) [61]. It appears that the use of LAMS < 15 mm was associated with greater clinical success and fewer adverse events than LAMS ≥ 15 mm [58].

A recent retrospective study compared patients drained by EUS-GBD (n = 41) and EUS-CDS (n = 37) after ERCP failure. The clinical success rates were similar (87.8% vs. 89.2%), as were technical success rates, but late morbidity (>24 h) was higher in the EUS-CDS group (21.6% vs. 7.3%). Notably, the mean CBD diameter was 13 mm in the EUS-GBD group (vs. 16 mm in the CDS group), demonstrating that this technique can be performed by non-experts when EUS-CDS is not feasible. Three patients in the EUS-GBD group later underwent pancreatic surgery, indicating that this technique does not contraindicate subsequent surgical management [62].

The promising results of this technique have led to its evaluation as a first-line treatment in distal malignant biliary obstruction compared to EUS-CDS. Seventy-seven patients were included in both groups after propensity score matching, with technical success rates > 95% and clinical success rates > 85%, showing no significant differences. There was no difference in adverse events (14% in both groups) and comparable event-free biliary survival [63]. Khoury’s meta-analysis demonstrated that EUS-GBD after ERCP failure was associated with a higher rate of adverse events than when used as first-line treatment (15.2% vs. 9%), explained by a higher incidence of post-ERCP pancreatitis [58].

Thus, this is a promising technique, technically more straightforward than EUS-CDS and especially EUS-HGS. A patent cystic duct is essential when a tumor is located more than 1 cm from the cystic duct [64,65], as evaluated with cross-sectional imaging and during EUS examination. Data from cholecystostomy suggest that a transduodenal approach is preferable to a transgastric route to reduce the risk of long-term adverse events and stent dysfunction [58]. There are currently no data on distal malignant biliary obstruction.

### 5.2. EUS-HGS Combined with Antegrade Stenting

A technique combining EUS-HGS with antegrade stenting (EUS-HGAS) of the papilla has recently been described for distal obstruction. The aim is to optimize biliary drainage and minimize RBO. Furthermore, one of the risks of EUS-HGS is peritonitis due to bile leakage; adding antegrade stenting may help direct bile flow toward the papilla.

A meta-analysis pooled data from 788 EUS-HGS and 295 EUS-HGAS procedures. The pooled technical success was 94% (95% CI: 92–96%) for EUS-HGS and 89% (95% CI: 83–93%) for EUS-HGAS, *p* = 0.00273. Pooled clinical success was 88% (CI: 84–91%) for EUS-HGS and 94% (95% CI: 89–97%) for EUS-HGAS, *p* = 0.0367. Pooled adverse events were 20% (95% CI: 16–25%) for EUS-HGS and 14% (95% CI: 9–20%) for EUS-HGAS, *p* = 0.072. Bile leakage had a 4% prevalence (95% CI: 2–9%) in the EUS-HGS group and 0% (95% CI: 0–5%) in the EUS-HGAS group. Of note, clinical efficacy was not homogeneously defined among the 25 studies included [66].

A recent study compared 81 EUS-HGS patients with 81 EUS-HGAS patients, demonstrating lower recurrence of biliary obstruction and a longer time to recurrence in the EUS-HGAS group (median, 194 days vs. 716 days; HR 0.050; 95% CI, 0.0066–0.37; *p* < 0.01) with no difference in overall survival [67]. This technique may therefore be beneficial for patients with prolonged survival.

## 6. Defining the Role of EUS-BD Techniques in the Management of Distal Malignant Biliary Obstruction in Pancreatic Cancer

### 6.1. Guidelines

Current European and American guidelines position EUS-BD as a second-line option after failed ERCP or in cases of inaccessible papilla, either due to tumor invasion of the duodenum or previous surgical alteration (such as a bypass) [68,69]. Historically, these situations were managed with PTBD, but data now show that this approach is associated with higher morbidity compared to EUS-guided drainage and should be reserved for hemodynamically unstable patients or those ineligible for general anesthesia [68]. The systematic review of the American Society of Gastrointestinal Endoscopy found that EUS-BD was associated with a higher rate of clinical success than PTBD (OR, 2.53; 95% CI, 1.22–5.28 in observational studies—though this was not seen in RCTs), a lower complication rate (OR 0.26 to 0.29 depending on the study type), fewer reinterventions, and shorter hospital stays (1.54 vs. 15.68 days, *p* < 0.05). There was no significant difference in terms of technical success and 30-day mortality [68]. These findings are confirmed by other recent meta-analyses [70,71]. Quality of life with PTBD is low, mainly due to external drains that may migrate, become obstructed, or kink.

### 6.2. EUS-BD as First-Line Therapy?

Several studies have evaluated the role of EUS-BD as a first-line approach compared to ERCP, even when ERCP is feasible. The ELEMENT trial is a randomized clinical study comparing EUS-CDS and ERCP as initial treatments for distal malignant biliary obstruction caused by borderline to unresectable periampullary cancers. The procedure time was shorter with EUS-CDS than with ERCP (14 vs. 23.1 min), with no difference in clinical success. EUS-CDS demonstrated non-inferiority in technical success compared to ERCP. There was no difference in stent dysfunction or adverse event rates [72]. Notably, no cases of acute pancreatitis were described in the EUS-CDS group. Another randomized trial (DRA-MBO) also compared EUS-CDS and ERCP for unresectable distal malignant obstruction [73]. In this trial, technical success was higher in the EUS-CDS arm compared to ERCP (96% vs. 76%), and the procedure time was again shorter (10 vs. 25 min), but there was no difference in clinical success, complication rates, or mortality.

These two trials were included in a meta-analysis of six trials involving 577 patients, comparing first-line EUS-BD—most receiving EUS-CDS—and ERCP. There was no significant difference in technical success rates (93.0%, 95% CI 89.3–95.5 in EUS-BD vs. 88.0%, 95% CI 79.1–93.4 in ERCP) or clinical success rates (89.0%, 95% CI 80.9–93.9 vs. 88.0%, 95% CI 83.3–91.5, respectively). The duration of biliary patency was not statistically different between groups (217.19 days, 95% CI 111.52–422.97 for EUS-BD vs. 210.66 days, 95% CI 108.49–409.02 for ERCP). The EUS-BD group had a lower reintervention rate than the ERCP group (9.75%, 95% CI 6.00–15.45 vs. 12.68%, 95% CI 5.39–27.00; relative risk 0.57, 95% CI 0.37–0.88; *p* = 0.01). There was no statistically significant difference in adverse events, but the pancreatitis rate was lower with EUS-BD than ERCP (0% vs. 5.83%, 95% CI 2.46–13.1; RR 0.15, 95% CI 0.03–0.66; *p* = 0.01). The rate of stent overgrowth was also lower in the EUS-BD group (2.63%, 95% CI 1.10–6.17 vs. 9.97%, 95% CI 4.00–22.72). The length of hospital stay was shorter in the EUS-BD group, and there was a significant trend toward shorter procedure time with EUS-BD compared to ERCP. When comparing patients who underwent EUS-CDS with LAMSs to those treated with ERCP, the technical success rates were higher and the procedure times were shorter in the LAMS-CDS group [74]. Another meta-analysis also confirmed shorter procedure times and higher technical success rates with LAMSs versus ERCP [13].

One of the main advantages of EUS-BD is the elimination of pancreatitis risk, a potentially severe complication that can alter patient prognosis. It also enables drainage distal to the tumor, reducing the risk of stent dysfunction from tumor ingrowth. For EUS-CDS with LAMSs, performing the procedure during the same diagnostic EUS session simplifies logistics and reduces overall procedure time. Thus, there is a growing trend toward EUS-guided management of distal malignant biliary obstruction as first-line therapy, particularly via EUS-CDS, marking a significant paradigm shift. However, as we already mentioned, not all cases of distal malignant biliary obstruction are eligible for this technique. The technical requirements for EUS-CDS are CBD dilation > 15 mm and the absence of vascular interposition or cystic duct interposition.

### 6.3. Specific Situations

#### 6.3.1. EUS-BD Before Surgery?

Most studies evaluating EUS-CDS involve patients without surgical plans. When surgery is considered, there is hesitation due to concerns about duodenal and biliary wall perforation. However, initial data are reassuring, and EUS-CDS does not appear to compromise surgical outcomes and may even offer preoperative benefits. A recent multicenter retrospective study examined the impact of LAMSs after EUS-CDS on subsequent surgery: A total of 937 patients were included (42 in the EUS-CDS group and 895 in the ERCP group). EUS-CDS was performed as a second-line procedure after a failed ERCP in 59.5% of cases. ERCP was performed with an FC-SEMS in 78% of cases. Postoperative complications were not significantly different between groups: 19% for EUS-CDS vs. 32.6% for ERCP (RR 0.50; 95% CI 0.23–1.07). Total complications were similar in both groups. There was no difference in postoperative bile leak or pancreatic fistula. The interval between drainage and surgery was significantly shorter in the EUS-CDS group (median 32 vs. 41 days; *p* = 0.01), as was the operative time (median 309 vs. 349 min; *p* = 0.002). These results remained valid after propensity score adjustment. A survey of surgeons who performed 29 pancreaticoduodenectomies after EUS-CDS found that surgery was considered not (45%), slightly (31%), clearly (17%), or severely (7%) more complex because of the EUS-CDS [75].

This study complements a recent French study that found fewer complications and a shorter hospital stay in the EUS-CDS group compared to ERCP [76]. In a smaller study involving 21 patients, pancreaticoduodenectomy was feasible in all cases [77]. While some conclusions differ, overall, there are no “red flags” contraindicating EUS-CDS in this context.

The effect of EUS-HGS on surgery is less well known, but its distance from the operative field suggests only a moderate impact. In a retrospective study of 81 patients who underwent pancreatoduodenectomy with preoperative biliary drainage (57 with ERCP stents and 20 with EUS-HGS), there was no significant difference in postoperative complications or length of hospital stay. There was also no difference in the rate of intraoperative peritoneal cytology positivity, and intraoperative bile cultures were less frequently positive in the EUS-HGS group compared with the ERCP stent group [78]. Overall, EUS-HGS does not appear to have a negative impact on surgical management, although available data remain limited.

#### 6.3.2. EUS-BD in Case of Concomitant Duodenal Obstruction

In cases of duodenal obstruction associated with biliary obstruction, the standard treatment was previously duodenal stenting to reach the papilla and retrograde biliary stent placement. In certain cases, a biliary stent may be placed after duodenal stenting. These strategies remain complex [79]. EUS-guided gastroenterostomy (EUS-GES) has changed the management of these patients. This technique involves deploying a LAMS between the stomach and the duodenum downstream of the cancer-related obstruction. This technique has been evaluated and compared with duodenal stenting in a multicenter randomized controlled clinical trial. It shows similar technical, clinical, and adverse event rates, but with a reduced frequency of reintervention with EUS-GES and better patient-reported eating habits compared with duodenal stenting [80]. Surgery is also an option. The CABRIOLET study is a retrospective analysis of different drainage options in patients with both biliary and duodenal obstruction. Six strategies were analyzed, including combinations of duodenal stenting with biliary stenting, EUS-CDS, EUS-HGS, and EUS-GES, along with these procedures. Ninety-three patients were included; the primary endpoint was the rate of dysfunction (recurrence of biliary or duodenal obstruction). The combination of enteral stenting plus EUS-CDS had the highest rate of clinical failure (40%) and the highest dysfunction rate (80%). There were no dysfunctions when EUS-HGS was combined with EUS-GES. In multivariate analysis, the presence of distal stenosis beyond the papilla and the combination of EUS-CDS with enteral stenting were identified as risk factors for dysfunction. A main limitation of this study is the small number of patients in each group (one in enteral stenting + EUS-HGS and six in EUS-GES + EUS-HGS), meaning the results should be interpreted cautiously, but they suggest better efficacy of the EUS-GES + HGS combination in double obstruction. In this study, the median time between the alimentary and biliary procedures was 31 days (range: 5–68 days) [81].

Another retrospective study with a small sample (39 patients) found EUS-HGS to be superior to EUS-CDS in cases of concomitant duodenal obstruction, with increased patency and a higher rate of complications (especially biliary reflux) seen with EUS-CDS [82]. Standard biliary stents were used in this study (not LAMS). The CABRIOLET PRO study prospectively evaluated biliary drainage in patients who had already undergone EUS-GES, comparing a group managed with EUS-CDS (7 patients) to EUS-HGS (13 patients). Technical and clinical success rates were similar, with a shorter procedure time in the EUS-CDS group (12.5 vs. 50 min). However, there was a higher adverse event rate in the EUS-CDS group (42.9% vs. 7.7%, *p* = 0.067), including one case of cholangitis-related death and a higher rate of biliary dysfunction due to food impaction [19].

While there is a lack of large prospective studies, current data suggest that EUS-HGS is preferable for biliary obstruction in the context of duodenal obstruction compared with other techniques.

#### 6.3.3. Proposed Management Algorithm of Distal Malignant Biliary Obstruction in Pancreatic Cancer

The proposed management algorithm for distal malignant biliary obstruction in the context of pancreatic cancer is based on recent advances in interventional endoscopic ultrasound while considering clinical complexity and patient anatomical specifics (Figure 2). ERCP remains the first-line recommendation when papillary access is possible, especially in the absence of duodenal obstruction or altered digestive anatomy. However, some data suggest that EUS-BD and particularly EUS-CDS could also be used as first-line therapy.

If ERCP fails, EUS-BD becomes the preferred alternative, with a preference for EUS-CDS, provided that technical conditions are met. In the case of an inaccessible papilla due to tumor infiltration and in the absence of duodenal stenosis, EUS-CDS is the technique of choice. In cases of tumor infiltration and duodenal obstruction, EUS-HGS combined with a digestive bypass (EUS-GES or duodenal stent) is advised, with recent data suggesting that EUS-GES could be preferable. EUS-HGS is the technique of choice in the case of an inaccessible papilla due to altered anatomy. The positioning of EUS-GBD and EUS-HGAS remains to be evaluated. Finally, in certain specific situations (failure of endoscopic techniques or inaccessible duodenum), PTBD remains a last resort.

## 7. Conclusions

Recent advances in EUS-BD have revolutionized the management of distal malignant biliary obstruction in the context of pancreatic cancer, and the development of EUS-CDS and EUS-HGS techniques has optimized care for these patients when ERCP fails or is not feasible. Recent studies increasingly support considering these techniques as first-line options rather than ERCP. Emerging strategies, such as EUS-HGAS or EUS-GBD, offer additional therapeutic alternatives. Further work is still needed to reduce the relatively high rate of stent dysfunction, clarify the appropriate use of each technique, and assess their impact prior to potential oncological surgery.

## Figures and Tables

**Figure 1 cancers-17-03428-f001:**
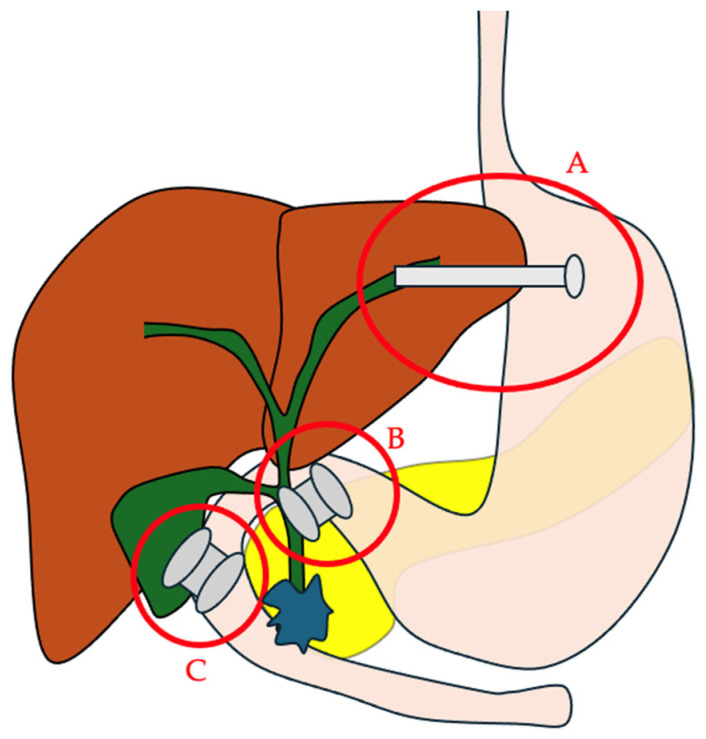
The main EUS-guided biliary drainage routes. A. EUS–hepaticogastrostomy; B. EUS–choledochoduodenostomy; C. EUS–gallbladder drainage.

**Figure 2 cancers-17-03428-f002:**
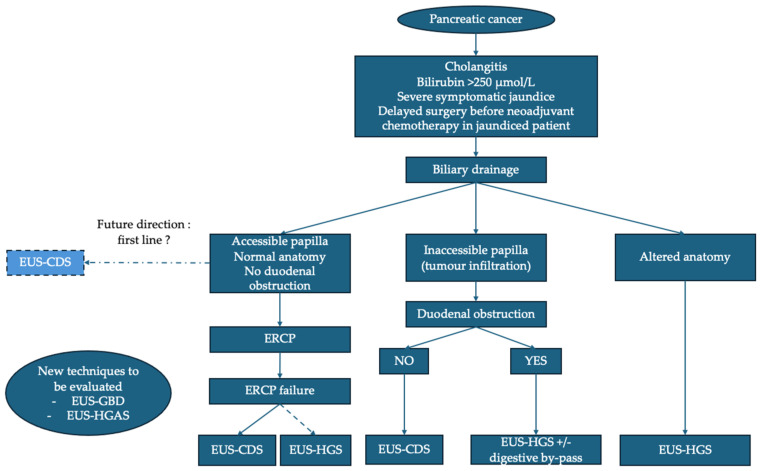
Proposed algorithm for management of distal malignant biliary obstruction in pancreatic cancer. ERCP: endoscopic retrograde cholangiopancreatography; EUS: endoscopic ultrasound; EUS-CDS: EUS–choledochoduodenostomy; EUS-GBD: EUS–gallbladder drainage; EUS-HGS: EUS–hepaticogastrostomy; EUS-HGAS: EUS–hepaticogastrostomy with antegrade stenting.

**Table 1 cancers-17-03428-t001:** Misposition classified according to LAMS type and endoscopic management.

Misposition Type	Description	Endoscopic Management
Type 1	Stent placed only in the CBD with the digestive flange mispositioned	Placement of a FC-SEMS and/or pigtails
Type 2	Stent located only in the digestive tract, with the biliary flange mispositioned	Guidewire placement through the LAMS catheter in place and deployment of a fully covered SEMS (FC-SEMS) Immediate placement of a new LAMS
Type 3	LAMS deployment failure	Deploying an FC-SEMS
Type 4	Catheter traverses the CBD from one side to the other	News EUS-CDS with FC-SEMS over a guidewire Repositioning the catheter with a new LAMS for guidewire insertion
Type 5	Bleeding (5a)	Clip or hemostatic stenting
	Complex technical failures and misplacement (5b)	

CBD: common bile duct; EUS: endoscopic ultrasound; EUS-CDS: EUS–choledochoduodenostomy; FC-SEMS: fully covered self-expanding metal stent; LAMS: lumen-apposing metal stent.

**Table 2 cancers-17-03428-t002:** LAMS dysfunction types.

LAMS Dysfunction Type	Description
Type 1	Sump syndrome (accumulation of bile and debris in the reservoir, leading to cholangitis)
Type 2a	Stone/sludge impaction
Type 2b	Food impaction
Type 3a	LAMS invasion/compression on the biliary side
Type 3b	LAMS invasion/compression on the duodenal side
Type 4	LAMS migration
Type 5	GOO

GOO: gastric outlet obstruction; LAMS: lumen-apposing metal stent.

**Table 3 cancers-17-03428-t003:** The main characteristics of EUS-CDS and EUS-HGS.

Characteristics	EUS-HGS	EUS-CDS
Principle	Creation of a connection between the dilated left bile ducts and the gastric cavity	Creation of communication between the duodenum and the common bile duct to bypass the tumoral obstruction
Main indications	Distal biliary obstruction with duodenal stenosis or altered anatomy	Distal biliary obstruction without significant duodenal stenosis
Access	Gastric (segments II or III of the liver)	Duodenal bulb (dilated CBD)
Type of stent	Dedicated partially covered SEMS	Mainly LAMS
Recommended technical criteria	Bile duct diameter > 5 mm; intraparenchymal portion 2.5–3 cm	CBD diameter > 15 mm (non-experts) or >12 mm (experts)
Technical success rate	>90%	>90%
Clinical success rate	>85%	>85%
Complication rate	Up to 30%	Up to 20%
Learning curve	Long, requires an expert operator	More accessible to novice operators
Procedure duration	Longer than EUS-CDS	Shorter than EUS-HGS

CBD: common bile duct; EUS: endoscopic ultrasound; EUS-CDS: EUS–choledochoduodenostomy; SEMS: self-expanding metal stent; LAMS: lumen-apposing metal stent.

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
