# Peer review of "Advances in EUS-Guided Biliary Drainage for the Management of Pancreatic Cancer"

_cancers, 2025, doi:10.3390/cancers17213428_

Round 1
Reviewer 1 Report
Comments and Suggestions for Authors
This was well written review about the EUS-BD/A (EUS-guided biliary drainage/anastomosis) for pancreatic cancer. We can learn the current status of this procedure in the management of pancreatic cancer with obstructive jaundice.
Major
- Interventional EUS is relatively new procedure, and the terminology is not established. Recently, an article about the terminology of I-EUS was published and articles about I-EUS should follow these definitions.
Isayama H, Nakai Y, Matsuda K, et al. Proposal of classification and terminology of interventional endoscopic ultrasonography/endosonography. Dig Endosc. 2025;37:5-17.
For examples: EUS-BD→EUS-BD/A, Fistula→anastomosis or tract
- Recently, some articles were reported about the preoperative EUS-HGS. Please discuss this new concept in your review.
- The advanced cases with massive ascites are the issue to be discussed as well
Author Response
- Comments 1: Interventional EUS is relatively new procedure, and the terminology is not established. Recently, an article about the terminology of I-EUS was published and articles about I-EUS should follow these definitions.
- Response 1: Thank you for pointing this out. We have replaced the term “fistulous tract” with “anastomosis” throughout the manuscript. These changes are highlighted in red on lines 92, 235, and 239. We have also revised the abbreviation for endoscopic ultrasound-guided gastroenterostomy to align with the recommended nomenclature (from EUS-GE to EUS-GES). These modifications appear on lines 520, 525, 529, 534, 537, 538, 546, 570 and 571. Other abbreviations, including EUS-BD, are consistent with those used in the reference you suggested.
- Comments 2: Recently, some articles were reported about the preoperative EUS-HGS. Please discuss this new concept in your review.
- Response 2: Agree. We have revised section 6.c.i to emphasize this point. The modifications are highlighted in red on page 14, line 505. The title of the section has also been modified to “EUS-BD before surgery?” on page 13, line 480.
- Comments 3 : The advanced cases with massive ascites are the issue to be discussed as well
- Response 3 : Thank you for highlighting this specific point. We have added details in the technical sections of both the EUS-CDS and EUS-HGS parts, emphasizing that the presence of ascites requires prior drainage and that these procedures should be performed by an experienced endoscopist, particularly in the case of EUS-HGS. The corresponding changes are highlighted in red on page 3, line 100 and on page 7, line 244.
Reviewer 2 Report
Comments and Suggestions for Authors
This review provides a comprehensive summary of the latest advances in endoscopic ultrasound-guided biliary drainage (EUS-BD) for the management of distal biliary obstruction caused by pancreatic cancer. The manuscript is well-structured, logically organized, and adequately referenced, offering high clinical relevance and educational value. However, several minor issues should be addressed before the manuscript can be considered for publication.
- The Abstractand Simple Summary are highly repetitive and do not sufficiently highlight the unique value or perspective of this review. The authors are advised to differentiate their content, emphasizing the review’s originality and clinical significance.
- The manuscript includes extensive listings of meta-analyses but lacks discussion of heterogeneity, limitations, and clinical generalizability among the included studies. It is recommended to provide critical commentary weaknesses and evidence gaps of the current literature.
- The technical descriptions in Sections 2–6 are overly detailed and lengthy, which may lead to reader fatigue. Consider streamlining procedural details and strengthening the analytical synthesis.
- Please incorporate discussion of the most recent randomized controlled trials, such as SCORPION-IIpand CABRIOLET-Pro, and highlight how these studies have influenced the evidence level supporting EUS-BD as a potential first-line approach.
- It is recommended to add a “Summary Table of Clinical Outcomes of EUS-BD Techniques,”summarizing technical and clinical success rates, adverse events, and learning curve data for easier comparison.
- Inconsistency was found in terminology — “EUS-HGAS” vs. “EUS-HAGS” (Page 12, Line 492). Please verify and standardize the abbreviation throughout the manuscript.
- Line 183: “This technique, first described in 2003 by Giovannini et al.” However, reference [32] actually refers to 2001, not 2003. Please check and correct this discrepancy.
- Line 231: “RBO occurred in 33% (36) of cases.” It is unclear what “36” represents. If it refers to the number of cases, please specify it clearly (e.g., n = 36).
Author Response
- Comments 1 : The Abstract and Simple Summary are highly repetitive and do not sufficiently highlight the unique value or perspective of this review. The authors are advised to differentiate their content, emphasizing the review’s originality and clinical significance.
- Response 1 : We sincerely thank the reviewer for this valuable comment. We have revised the Simple Summary to reduce redundancy and to better emphasize the originality and clinical relevance of our review. The updated versions now highlight the unique perspective of our work and its contribution to current clinical practice. These modifications are highlighted in red on page 1, lines 9.
- Comments 2 : The manuscript includes extensive listings of meta-analyses but lacks discussion of heterogeneity, limitations, and clinical generalizability among the included studies. It is recommended to provide critical commentary weaknesses and evidence gaps of the current literature.
- Response 2 : We appreciate the reviewer’s insightful comment. Meta-analyses were used in our review to obtain data that most closely reflect real-world outcomes for the main EUS-BD techniques. We have now added comments regarding the quality and limitations of these meta-analyses to provide a more critical perspective on the current evidence. These additions can be found on in red page 11, line 362, and page 12, line 406.
- Comments 3 : The technical descriptions in Sections 2–6 are overly detailed and lengthy, which may lead to reader fatigue. Consider streamlining procedural details and strengthening the analytical synthesis.
- Response 3 : We thank the reviewer for this comment, which helps improve the readability of the manuscript. We have removed technical details from the “Principles” sections of both the EUS-CDS and EUS-HGS parts (pages 2, 3, and 7). It should be noted that some elements had to be added following requests from other reviewers, including details on the type of LAMS (page 3, line 83) and on ascites management (page 3, line 100 and page 7, line 244).
- Comments 4 : Please incorporate discussion of the most recent randomized controlled trials, such as SCORPION-IIpand CABRIOLET-Pro, and highlight how these studies have influenced the evidence level supporting EUS-BD as a potential first-line approach.
- Response 4 : We thank the reviewer for this helpful suggestion. The name of the SCORPION-IIp trial, which evaluated the routine placement of a fully covered self-expandable metal stent (FC-SEMS) within the LAMS to reduce the risk of stent dysfunction, has now been added in red on page 6, line 201, for better visibility. The CABRIOLET-Pro study is already mentioned in our review and discusses the optimal drainage approach in cases of combined biliary and duodenal obstruction. In addition, randomized controlled trials evaluating EUS-BD as a potential first-line approach, including the ELEMENT and DRA-MBO trials, are discussed on page 12 lines 437 and page 13 line 444.
- Comments 5 : It is recommended to add a “Summary Table of Clinical Outcomes of EUS-BD Techniques,”summarizing technical and clinical success rates, adverse events, and learning curve data for easier comparison.
- Response 5 : We thank the reviewer for this constructive suggestion. Creating an additional summary table would be somewhat redundant, as Table 3 already provides an overview of the clinical outcomes of EUS-HGS and EUS-CDS, which we would like to maintain. The purpose of our review was to emphasize that these two techniques currently represent the main options to be discussed for distal biliary obstruction, and Table 3 positions each technique in relation to the other. We did not include EUS-RV and EUS-AS in our table, as these methods are now rarely used. Similarly, EUS-GBD and EUS-HGAS were not added, as these are relatively recent techniques for biliary drainage that require further investigation and cannot yet be compared on the same level as EUS-HGS and EUS-CDS. We hope the reviewer will understand our rationale for this decision.
- Comments 6 : Inconsistency was found in terminology — “EUS-HGAS” vs. “EUS-HAGS” (Page 12, Line 492). Please verify and standardize the abbreviation throughout the manuscript.
- Response 6 : We thank the reviewer for this careful observation. The inconsistency has been corrected on page 16, line 588.
- Comments 7 : Line 183: “This technique, first described in 2003 by Giovannini et al.” However, reference [32] actually refers to 2001, not 2003. Please check and correct this discrepancy.
- Response 7 : We thank the reviewer for this helpful comment. The discrepancy has been corrected, and the reference has been updated accordingly on page 20, line 742.
- Comments 8 : Line 231: “RBO occurred in 33% (36) of cases.” It is unclear what “36” represents. If it refers to the number of cases, please specify it clearly (e.g., n = 36).
- Response 8 : We thank the reviewer for noticing this point. The sentence has been modified to clearly specify the number of cases, as suggested. The revision can be found on page 8, line 282.
Reviewer 3 Report
Comments and Suggestions for Authors
The article provides a comprehensive review of the use EUS drainage in the management of distal malignancy.
I only recommend taking into consideration the following:
Please check the bilirubin expression level. Within the third page you use bilirubin with mg/dl
Why is EUS gallbladder drainage is considered an emerging technique?
Please include a discussion of the stents available: hot axios vs hot spaxus, Zstent.
Author Response
- Comments 1 : Please check the bilirubin expression level. Within the third page you use bilirubin with mg/dl
- Response 1 : We sincerely thank the reviewer for this observation. The bilirubin unit has been corrected for consistency and is now highlighted in red on page 3, line 108.
- Comments 2 : Why is EUS gallbladder drainage is considered an emerging technique?
- Response 2 : EUS-guided gallbladder drainage has been described for several years; however, using the gallbladder as a conduit for biliary decompression remains a relatively new and less frequently reported approach. Given the limited but promising data available, we have therefore considered this technique as an emerging option. To clarify this point, we have modified the title of the corresponding section on page 10, line 349.
- Comments 3 : Please include a discussion of the stents available: hot axios vs hot spaxus, Zstent.
- Response 3 : We agree with the reviewer’s comment. A sentence has been added to clarify the types of stents that can be used for EUS-CDS, emphasizing that the Hot Axios™, Hot Spaxus™, and Z-Stent™ are among the available options, with Axios™ and Spaxus™ being the most frequently employed. This modification has been added and highlighted in red on page 3, line 83.
Round 2
Reviewer 2 Report
Comments and Suggestions for Authors
The authors have adequately addressed all the points I raised in the previous review. The revisions made to the manuscript are appropriate and have significantly improved the clarity and quality of the work. I therefore recommend acceptance of the manuscript in its present form.